# A Cross-Domain Method for Customer Lifetime Value Prediction in Supply Chain Platform

## ABSTRACT

Accurate Customer Lifetime Value (LTV) predictions are crucial for customer relationship management, especially in Supply Chain Platforms (SCP), which involve effectively managing the service resources in business decision-making. Previous LTV prediction methods usually rely on ample historical customer data, which is not available in the early stages of a customer's lifecycle. It makes the modeling of the historical customer data a difficult task due to the data sparsity. Besides, the long-tail distribution of customer LTV also brings new challenges to the prediction of LTV. To tackle the above issues, we propose **CDLtvS**, a novel **C**ross **D**omain method for customer **L**ife**t**ime **v**alue prediction in **S**CP. It leverages rich cross-domain information from upstream platforms to enhance LTV predictions in downstream platforms. Firstly, CDLtvS pre-trains the customer representations by an LTV modeling framework named LtvS in source and target domains separately. Specifically, LtvS incorporates the Expert Mask Network (ExMN), which not only effectively models the long-tail distribution of LTV in single-domain but also resolves cross-domain learning model bias resulting from this distribution. Then, the various-level alignment mechanism is introduced to keep the consistency of knowledge transferring from source to target domains on both sparse and non-sparse data. Comprehensive experiments on real-world data from JD, one of the world's largest supply chain platforms, demonstrate that CDLtvS achieves a normalized mean average error of 0.3378 in LTV prediction, outperforming 16.3% to the baseline. Additionally, the improvements of ≥2.3% across various data sparsity levels (0% – 80%) provide valuable insights into cross-domain LTV modeling.

## CCS CONCEPTS

• **Information systems → Information systems applications**.

## KEYWORDS

Lifetime Value Prediction, Supply Chain Platform, Cross-Domain Knowledge, Economics

**ACM Reference Format:**
Anonymous Author(s). 2018. A Cross-Domain Method for Customer Lifetime Value Prediction in Supply Chain Platform. In *Proceedings of The ACM Web Conference (WWW'24)*. ACM, New York, NY, USA, 9 pages. https://doi.org/XXXXXXX.XXXXXXX

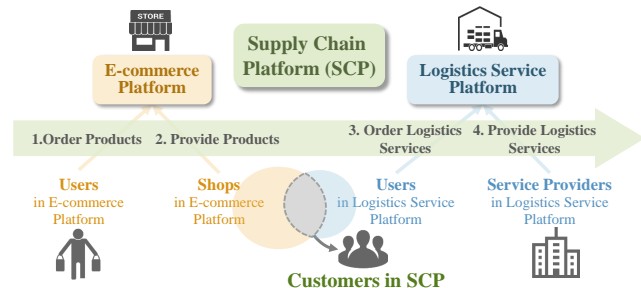

**Figure 1: A typical Supply Chain Platform (SCP) working flow consists of four steps: 1, the e-commerce platform receives orders from end users and 2, notifies the customers to prepare the product accordingly; in Step 3, the customers order logistics services (e.g., shipment), which are processed by the logistics service platform in Step 4.**

## 1 INTRODUCTION

In the current era of web-based business and digital economics, many companies have been leveraging artificial intelligence technologies for data-driven precision in customer relationship management (CRM), thereby effectively enhancing both company revenue and customer satisfaction [3]. An essential factor in this process is Customer LifeTime Value (LTV), which plays a significant role in the assessment and decision-making procedures of CRM [1, 11]. Especially in Supply chain platforms (SCP), accurate prediction of LTV is crucial for companies to guide the rational allocation of marketing resources and incentive special customer segments [21].

LTV prediction is a time series forecasting problem that aims to predict a customer's future consumption value based on the historical consumption series. While the recent development of deep neural networks has significantly improved LTV prediction performance [5, 16], these models typically require ample historical customer data, which is impractical, especially in the early stages of the customer lifecycle. To address the similar data sparsity issue in recommendation systems, existing work proposes cross-domain recommendation (CDR) [28], a technique that employs rich domain information (source domain) to enhance recommendation accuracy in sparse domains (target domains). In the context of SCP, characterized by significant customer overlap between upstream and downstream platforms, cross-domain recommendation methods can be extended to alleviate the data sparsity issue by leveraging abundant cross-domain information from the upstream platform to benefit the downstream platform.

This paper focuses on the cross-domain customer LTV prediction problem within the SCP scenario, comprising two linked components: the e-commerce platform and the logistic platform. The e-commerce platform is basically for online shopping involving interactions between users and customers (i.e., online shops), while the logistic platform is for logistic service transactions between

customers and service providers (e.g., logistic companies). We note that the customers on the e-commerce platform also serve as the "users" on the logistics service platform. Subsequently, end users order products on e-commerce platforms, where the customers [1] prepare the products, which are further fed into the logistic platform for shipment, following SCP working flow, as depicted in Figure 1. The SCP customers' logistic demand is influenced by the business generated on the e-commerce platform. Leveraging cross-domain information from the e-commerce platform (source domain) can help mine more customer information and enhance LTV prediction on the logistics service platform (target domain).

However, LTV prediction in SCP remains a challenging task due to several factors: (1) *The long-tail distribution of LTV is challenging to model.* Figure 2 illustrates that the LTV distributions in both the source and target domains are imbalanced. In stochastic gradient optimization, the large gradient vectors generated by high LTV samples potentially impact model convergence and stability. (2) *Long-tail data lead to biased cross-domain information transfer.* The prevalence of head samples can hinder the adequate modeling of high-value customers in cross-domain knowledge learning. (3) *Learning effective source domain knowledge transfer patterns is a challenge*, particularly when building upon limited non-sparse customer samples to improve predictions for sparse customers. Straightforward cross-domain presentation aggregation with inherently noisy doesn't effectively refine the migration of cross-domain information. Current methods mainly focus on LTV problems related to customer representation learning [26, 29] or addressing long-tail data distribution modeling [16, 25]. However, these methods rely on the assumption of having ample historical customer data within a single domain. Although many efforts have been made to solve the data sparsity problem in cross-domain recommendation [17, 32], the long tail distribution of LTV is still an open issue hard to solve.

To address the above challenges, we introduce **CDLtvS**, a **C**ross-**D**omain method for customer **L**ife**t**ime **v**alue prediction in **S**upply chain platform. CDLtvS firstly pre-trains the personalized customer representations in the source and target domains separately with a single-domain modeling method LtvS, then jointly leverages the pre-trained representations to enhance the cross-domain LTV features for future value prediction. Specifically, (1), an *Expert Mask Network* (ExMN) is designed to segregate head and tail value data into distinct feature spaces using unique vector masks, addressing the long-tail distribution data modeling challenge. (2), ExMN also mitigates the issue of biased cross-domain knowledge transfer by utilizing pre-trained single-domain representations enriched with distribution information. (3), we introduce *cross-domain knowledge alignment at various levels* to finely tune the transfer of cross-domain information. Overall, the contributions of this paper include:

- We propose CDLtvS, a novel method that tackles data sparsity issues in LTV prediction by leveraging cross-domain information in the SCP. To the best of our knowledge, we are the first to address cross-domain LTV prediction in SCP.
- To effectively model the long-tail LTV distribution, CDLtvS introduces an Expert Mask Network in single-domain representation modeling. ExMN's design also adeptly addresses the model bias problem in cross-domain learning.

---

[1] In this paper, we refer to business customers as customers for short.

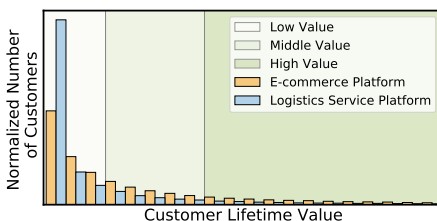

**Figure 2: The long-tail LTV distribution in the upstream and downstream industry platforms of this SCP.**

- CDLtvS further enhances cross-domain knowledge transfer for both sparse and non-sparse data through a various-level alignment mechanism.
- Extensive experiments are conducted using a real-world dataset, including 5,000 customers collected from JD.com, one of the world's largest supply chain platforms. The results outperform the baseline methods by 2.3%, 11.5%, 16.3%, and 14.4% in normalized mean average error under various data sparsity levels (0% to 80%), showing the effectiveness of our CDLtvS approach for LTV prediction.

## 2 RELATED WORK

### 2.1 LTV Prediction

LTV prediction has evolved over time, driven by the growing recognition of its significance in marketing and customer relationship management. In the early stages, LTV prediction methods were constrained by data limitations and relied on probabilistic statistical models [7–9], such as the BTYD model and Pareto/NBD [8]. Recently, the emergence of large-scale e-commerce platforms accumulating vast amounts of data has facilitated the adoption of machine learning and deep neural network methods [5, 27], leading to more accurate LTV prediction. For example, Drachen et al. [6] utilizes a two-stage XGBoost to identify high-value customers and predict their monetary value from the social features. Xing et al. [26] focus on jointly learning temporal and structural customer representations based on historical customer behavior. Piao et al. [23] leverage social network graphs as auxiliary customer information to enhance the learning of structural customer representations. Wang et al. [25] and Li et al. [16] concentrate on modeling the long-tailed distributions of LTV. Wang et al. [25] assume that LTV follows a zero-inflated lognormal (ZILN) distribution with limitations in its applications, whereas Li et al. [16] propose a decomposition approach relying on prior knowledge of LTV distribution to address the issue of imbalanced sample distributions.

However, all the aforementioned methods utilize one-domain data, relying on the availability of abundant historical customer data for training, which is a limiting assumption in real-world scenarios. Customer historical information is often sparse in the early stages of the customer life cycle, and making LTV predictions at this stage is crucial for aiding companies in managing customer relationships.

### 2.2 Cross-domain Recommendation

Similar to LTV prediction, recommendation systems are faced with the data sparsity problem, which promote the emergence

and development of Cross-Domain Recommendation (CDR) [14, 17–19, 24, 32]. The core idea of CDR is to leverage information collected from other domains to alleviate the data sparsity problem in one domain [28].

One popular paradigm is to customize a mapping bridge function whose optimization objective is that the transformed user representation in the source domain generalizes well in the target domain, like EMCDR [20], PTUPCDR [32], and so on. The efficiency of this paradigm is damaged when applied to non-sparse data because of the loss of target domain information. Another group of CDR methods utilizes information from both the target domain and the source domain by introducing transfer learning techniques to transfer knowledge between domains. CoNet [13] enables dual knowledge transfer across domains by using cross-connections between feed-forward neural networks. DASL [17] takes into account sequential information during the recommendation process and designs dual-attention learning mechanism to transfer the knowledge contained in the history sequences. however, the lack of cross-domain alignment and the rough combination way limit their performance.

Our approach aligns with the second paradigm. We transfer the source domain knowledge and fuse cross-domain information with cross-domain alignment at different levels. However, to the best of our knowledge, no prior research has applied CDR techniques to the domain of LTV prediction.

## 3 PROBLEM DEFINITION

This paper studies the task of predicting the future Lifetime Value (LTV) of customers in the Supply Chain Platform (SCP). In SCP, we have an upstream industry platform, serving as the source domain, and a downstream industry platform, serving as the target domain. The customer of SCP is defined as the entity overlapping both service providers in the upstream industry platform and service users in the downstream industry platform. We represent the customer set of SCP as $C$. This set contains two customer types, one is $C_a$ possessing ample historical information in both source and target domains, while the other is $C_{na}$ with sparse historical information in the target domain. We denote the customer historical consumption series and the customer portrait features of the source domain as $H^{s,l}$ ($H^{s,l} = [h_0^{s,l}, h_1^{s,l}, ..., h_N^{s,l}]$, $N$ is the length of $H^{s,l}$) and $h^{s,p}$, and similarly $H^{t,l}$ and $h^{t,p}$ for the target domain.

Given the above notation, for a customer $c$ ($c \in \{C_a, C_{na}\}$), along with their historical consumption series $H^{s,l}$, $H^{t,l}$, and portrait features $h^{s,p}$, $h^{t,p}$, our objective is to forecast the cumulative LTV for the future $n$ days starting from the $(N+1)$-th day in the target domain, denoted as $v_c = \sum_{i=N+1}^{N+\Lambda N+1} h_{c,i}^{s,l}$.

## 4 PROPOSED METHOD

We propose a **C**ross **D**omain method for customer **L**ife**t**ime **v**alue prediction in **S**upply chain platform, named **CDLtvS**. The primary design objective is to solve the data sparse problem in LTV prediction and address the challenges posed by long-tail LTV data and cross-domain learning.

CDLtvS begins by designing a single-domain modeling method, named LtvS, which incorporates an Expert Mask Network referred to as ExMN with the pattern segmentation constraints to train the

domain representation in long-tail data. Furthermore, we introduce an effective source domain knowledge transfer method that incorporates cross-domain alignment, enabling cross-domain LTV prediction for both sparse and non-sparse data.

### 4.1 Overview of CDLtvS Framework

The CDLtvS framework, illustrated in Figure 3, comprises three modules designed to predict customer lifetime value in the supply chain platform:

- **Target Domain Representation (TDR)**. This module utilizes customer portrait features and consumption series in the target domain. It first encodes these two types of data separately and then combines the resulting embeddings using the ExMN network we propose. This process yields the hidden representation, denoted as $e^t$, for the customers in the target domain.
- **Source Domain Representation (SDR)**. Following the same structure as the TDR module, this module obtains the hidden representation of customers in the source domain, denoted as $e^s$.
- **Cross-domain LTV prediction (CDP)**. In this module, we transfer the source domain representation, $e^s$, into the vector $e^m$ through knowledge transferring. Subsequently, we fuse the vectors $e^m$ and $e^t$ with a low-dimensional alignment process to accomplish cross-domain LTV prediction.

The CDLtvS framework follows a two-stage training method. In Stage 1, we pre-train the TDR and SDR modules. This can be viewed as two single-domain LTV predictions for two domains independently, serving as our single-domain modeling method named **LtvS**. In Stage 2, we use the pre-trained parameters from Stage 1 and jointly train the cross-domain LTV prediction module along with the representation modules of the two domains.

We provide a detailed explanation of the single-domain modeling method **LtvS** in **TDR** and **SDR** in the following section 4.2, as well as the training method in **CDP** in section 4.3.

### 4.2 LtvS: Single-Domain Modeling

In this section, we take the pre-training process stage 1 in the source domain as an example to illustrate how LtvS optimizes the customer representations within a single domain, which is denoted as $e^s$ for the source domain and $e^t$ for the target domain.

*4.2.1 **Feature Encoder**.* In our single-domain pre-training method, LtvS, we employ DeepFM[10] to encode customer portrait features $h^{s,p}$ due to its capability to capture complex feature interactions, resulting in vector $e^{s,p} \in \mathbb{R}^h$ (where $h$ is the hidden size of embeddings). For consumption series $H^{s,l}$, we apply the Temporal Convolutional Network (TCN) [2] generating vector $e^{s,l} \in \mathbb{R}^h$ due to its capability to model temporal patterns at various scales. The combination of DeepFM and TCN enhances the richness of information extracted from both types of data.

*4.2.2 **ExMN Network**.* We design a novel network called ExMN (**Ex**pert **M**ask **N**etwork) to address the long-tail LTV distribution problem, aiming to distinguish customers of varying value levels. The fundamental concept behind ExMN involves the utilization of an expert router to assign distinct vector masks for data in the head and tail segments. It encourages the model to map the distinct

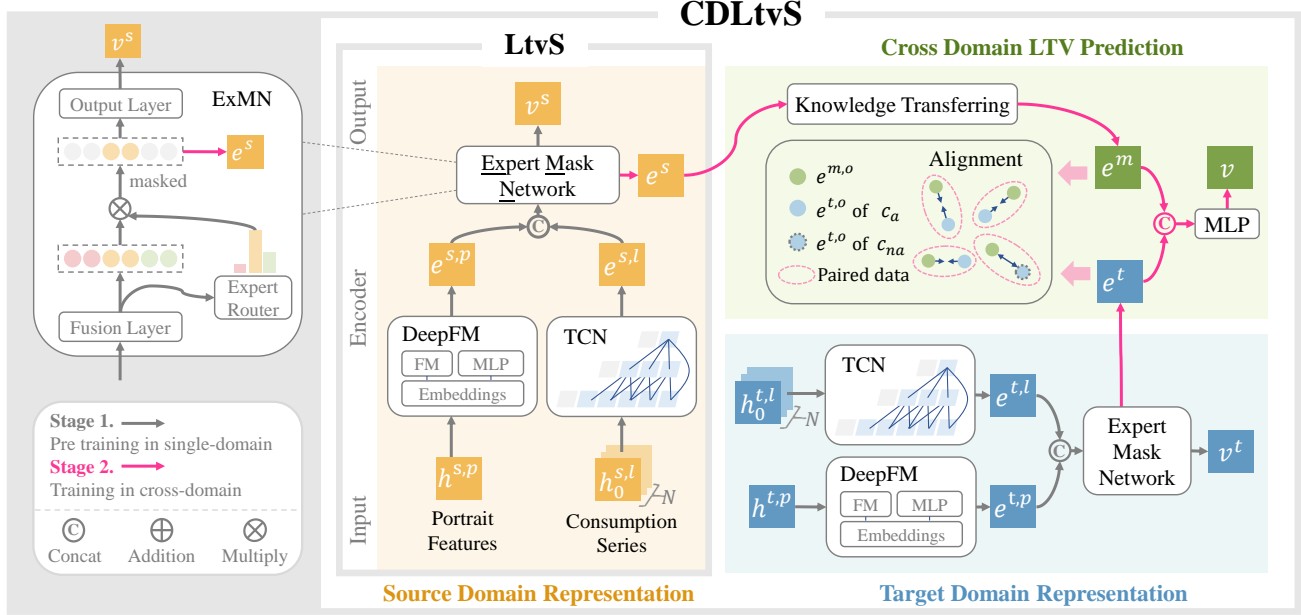

**Figure 3: The overall framework of CDLtvS. Initially, we acquire the hidden representations $e^t$ and $e^s$ from the pre-trained target domain and source domain representation modules using the single-domain modeling method LtvS. Subsequently, we employ these representations within both the target and source domains to facilitate cross-domain LTV prediction.**

feature segments within a high-level embedding space to several different sub-spaces to enhance a more balanced data distribution.

The structure of the ExMN Network is depicted on the left of Figure 3. Initially, the encoded feature embeddings are fused through a fusion layer to obtain the initial single-domain representation $\dot{e}^s$, given by the equation:

$$\dot{e}^s = ReLU(FFN_f([e^{s,p}, e^{s,l}])). \tag{1}$$

Here, $FFN_f$ contains a fully connected (FC) layer, and ReLU serves as the activation function.

To allocate distinct vector masks for different data, we employ an expert router network consisting of two FC layers, $FFN_{smax}$ and $FFN_g$, to calculate the routing probability $\dot{g}^s \in \mathbb{R}^K$. $K$ represents the number of experts, and each expert points to an independent non-mask location. The following show equations:

$$g^s = Softmax(\mathcal{T}(FFN_{smax}(\dot{g}^s), 1)) \tag{2}$$

$$\dot{g}^s = ReLU(FFN_g(\ddot{e}^s)) \tag{3}$$

$$\mathcal{T}(x, a) = \begin{cases} x_i & \text{if } x_i \text{ is in the top } a \text{ elements of } v, \\ -\infty & \text{otherwise.} \end{cases} \tag{4}$$

where $\mathcal{T}(x, a)$ represents the top-$a$ selection function, and in our work, we set $a = 1$ to assign a unique expert to each sample. This results in one element in $\dot{g}$ being equal to 1, and the rest equal to 0. It should be noted that this method can be extended to values of $a$ greater than 1, which allows mask combinations generated by the mixture of experts.

Next, we create $m = h \div k$ copies of each element in the vector $\dot{g}^s \in \mathbb{R}^k$ to obtain the masking vector $g^s \in \mathbb{R}^{h=k \times m}$. It's important to note that $h$ must be divisible by $k$. By dot product $g^s$ and $\dot{e}^s$, we

mask certain elements in $\dot{e}^s$, resulting in the new single-domain representation $e^s$. The process of masking helps create feature spaces for $e^s$ that are tailored to its expert routing. Furthermore, when $e^s$ serves as an input to the subsequent FC layers, it encourages the model to focus on different features for data with varying routing.

Finally, we feed $e^s$ to the output layer $FNN_o$ and get the final LTV prediction output, which can be expressed as:

$$\hat{v}^s = ReLU(FNN_o(e^s)). \tag{5}$$

*4.2.3 Optimization.* As discussed in Section 4.2.2, our aim is to achieve a more balanced data distribution for specific feature learning by utilizing the expert router to allocate distinct vector masks for customers of different value levels. To accomplish this, we associate expert routing with LTV distribution information through the following constraints:

$$\mathcal{L}_{ExMN} = \lambda_1 \cdot \mathcal{L}_{EC} + \lambda_2 \cdot \mathcal{L}_{EO} \tag{6}$$

where $\mathcal{L}_{EC}$ is the Expert Classification cross-entropy loss, and $\mathcal{L}_{EO}$ is the Expert Ordinal regression loss. $\lambda_1$ and $\lambda_2$ determine the contribution of each Expert loss.

The expert classification cross-entropy loss $\mathcal{L}_{EC}$ evaluates the model's capability to correctly categorize customer samples into their respective LTV levels or, in other words, how correctly they are assigned to routing experts. This assessment is made by penalizing the discrepancy between the predicted probability $\hat{p}_{EC}$, and the actual LTV level label $p_{EC}$. To obtain the actual label $p_{EC}$, the customer samples in a batch $\mathcal{B}$ are divided into $k$ customer groups based on their real LTV label $v^s$. And for the LTV level prediction $\hat{p}_{EC}$, we compute it by employing the FC layer $FNN_{smax}$ of the

router network with the middle vector $\dot{g}^s$ as input.

$$\mathcal{L}_{EC} = -\frac{1}{|\mathcal{B}|}\frac{1}{k}\sum_{c_i \in \mathcal{B}}\sum_{j=0}^{k-1} p_{EC}^{i,j} \cdot log(\hat{p}_{EC}^{i,j}) \tag{7}$$

$$\hat{p}_{EC} = Softmax(FFN_{smax}(\dot{g}^s)) \tag{8}$$

The expert ordinal Regression loss $\mathcal{L}_{EC}$ establishes the relative order relationship among routing experts, enhancing the accuracy of classification and ranking.

$$\mathcal{L}_{EO} = -\frac{1}{|\mathcal{B}|}\frac{1}{k}\sum_{c_i \in \mathcal{B}}(\sum_{j=0}^{m} log(\hat{p}_{EO}^{i,j}) + \sum_{j=m+1}^{k-1}(1 - log(\hat{p}_{EO}^{i,j})) \tag{9}$$

$$\hat{p}_{EO} = Sigmoid(FFN_{smoid}(\dot{g}^s)) \tag{10}$$

where $m$ represents the real label of the customer's LTV level ID, and $\hat{p}_{EO}^{j}$ indicates the probability that the LTV level is higher than expert $j$. We calculate this probability using an FC layer $FNN_{smoid}$ based on the intermediate vector $\dot{g}^s$.

Finally, we use Mean Average Error (MAE) as the LTV loss function, defined as:

$$\mathcal{L}_{LTV} = -\frac{1}{|\mathcal{B}|}\sum_{c_i \in \mathcal{B}}|v_i - \hat{v}_i|. \tag{11}$$

The total loss for the source domain pre-training in stage 1 is a combination of $\mathcal{L}_{LTV}$ and $\mathcal{L}_{ExMN}$:

$$\mathcal{L}^s = \mathcal{L}_{LTV}^s + \mathcal{L}_{ExMN}^s. \tag{12}$$

*4.2.4* **Discussion**. *(1) ExMN benefits long-tail distribution data modeling in single-domain.* With the structure of ExMN and the constraints imposed by the ExMN loss, our approach effectively addresses the challenge posed by the long-tail distribution in single-domain LTV modeling. It encourages the embedding vectors of customers with different LTV levels to be masked into different feature spaces, emphasizing different features for different LTV-level customers. This approach allows the model to be well-versed within different feature spaces corresponding to various LTV levels.

*(2) ExMN mitigates the issue of biased cross-domain information transfer in the subsequent cross-domain LTV prediction stage* with the distribution information brought by the pre-trained embedding vectors. The masked vectors create areas that highlight the tail of the distribution, which provides the model with more opportunities to learn and generalize effectively from these tail data, thereby mitigating the issue of model bias.

## 4.3 Cross-Domain LTV Prediction

In this section, we present the training process for cross-domain LTV prediction in the second stage. We begin by using a knowledge transferring module to transfer the source domain representation $e^s$ to a transferring knowledge embedding, denoted as $e^m$. Subsequently, we combine this transferring knowledge with the target domain representation, $e^t$, using a multi-layer perceptron (MLP) to yield cross-domain LTV predictions, as expressed by the following equations:

$$e^m = ReLU(FFN_{trans}(e^s) \tag{13}$$

$$\hat{v}^s = MLP([e^m, e^t]) \tag{14}$$

where $FFN_{trans}$ represents the fully connected layer of the knowledge transfer module. The $MLP$, composed of two fully connected

layers with ReLU activation functions, combines knowledge from both domains to facilitate LTV prediction.

*4.3.1* **Alignments at Various Levels**. To enhance cross-domain knowledge transfer for both sparse and non-sparse data, we introduce two alignment methods in a lower-dimensional space.

Initially, to ensure partial alignment of source domain transferring knowledge information with target domain information while preserving individual domain-specific details, we project the vectors requiring alignment into a lower-dimensional space. This projection is achieved using two matrices, denoted as $W^m$ and $W^t$, both having dimensions of $\mathbb{R}^{h \times d}$. We apply these matrices to transform the transferring knowledge representation $e^m$ and the target domain representation $e^t$, both initially in $\mathbb{R}^h$, into a lower-dimensional space. This transformation results in $e^{m,o}$ and $e^{t,o}$, which are now in $\mathbb{R}^d$. Importantly, this new embedding size $d$ is smaller than the original embedding size $h$.

Then, we propose alignment methods for paired non-sparse data. Our assumption is that the embedding $e^m$, learned through the transferring knowledge module, should resemble the target domain knowledge $e^s$ containing substantial information. Thus, we intend to mainly utilize transferring knowledge $e^m$ for LTV prediction when the customer's target domain information is insufficient. To achieve this, we introduce an alignment loss, denoted as $\mathcal{L}_{AP}$, for paired non-sparse data $C_a$. Following [30], we employ a contrastive learning approach InfoNCE [22], to enhance the similarity between the $e^{m,o}$ and $e^{t,o}$ vectors of the same customer which are treated as paired data. The loss is expressed as follows:

$$\mathcal{L}_{AP} = -\sum_{c_i \in \mathcal{B} \cap C_a} log \frac{exp(sim(e_i^{m,o}, e_i^{t,o})}{\sum_{c_j \in S_{c_i}} exp(sim(e_i^{m,o}, e_j^{t,o}))} \tag{15}$$

where $S_{c_i}$ represents the negative samples for customer $c_i$ in the batch $\mathcal{B}$. $sim(\cdot, \cdot)$ measures the cosine similarity between vectors.

Furthermore, we propose the alignment for data of the same type. We categorize customers into two types: $C_a$ with ample historical information and $C_{na}$ with limited historical information, which corresponds to non-sparse and sparse data, respectively. We apply positive alignment for sparse data and negative alignment for non-sparse data to distinguish these data types within the spatial representation. This approach allows the model to distinguish non-sparse and sparse data when learning knowledge fusion modes. The alignment loss $\mathcal{L}_{AT}$ for data of the same type is defined as:

$$\mathcal{L}_{AT} = \frac{1}{|\mathcal{B} \cap C_{na}|}\sum_{c_i \in \mathcal{B} \cap C_{na}} max(0, sim(e_i^{m,o}, e_i^{t,o}) - \eta)$$
$$+ \frac{1}{|\mathcal{B} \cap C_a|}\sum_{c_i \in \mathcal{B} \cap C_a}(1 - sim(e_i^{m,o}, e_i^{t,o})) \tag{16}$$

where $\eta$ is the similarity margin of sparse data.

*4.3.2* **Optimization**. In summary, the loss function $\mathcal{L}^c$ for our CDLtvS model in stage 2 is defined as follows.

$$\mathcal{L}^c = \mathcal{L}_{LTV} + \lambda_3 \cdot \mathcal{L}_{AP} + \lambda_4 \cdot \mathcal{L}_{AT} \tag{17}$$

where the strength of the alignment loss for paired non-sparse data is controlled by $\lambda_3$ and the strength of the alignment loss for data of the same type is controlled by $\lambda_4$.

# 5 EXPERIMENTS

## 5.1 Experimental Settings

*5.1.1 Datasets.* There is no publicly available dataset in LTV prediction for SCP. To validate our model's effectiveness, we gathered data from a real-world JD application, an SCP comprising (1) an E-commerce platform as the source domain, containing customer order number sequences and e-commerce portrait features, and (2) a logistics service platform as the target domain, containing customer waybill number sequences and logistic portrait features. In the JD dataset, we randomly sampled 5,000 customers in Guangzhou from August 10, 2020, to October 24, 2022. We segmented the customer historical sequence data into 30-day intervals, resulting in 88,577 samples for experiments on the future 30-day LTV prediction. For evaluation purposes, we divided the JD dataset into training, validation, and test sets in a ratio of 7:2:1.

*5.1.2 Evaluation Metrics.* In this paper, we evaluate the proposed framework using key metrics, including the Normalized Mean Average Error (NMAE), Normalized Rooted Mean Square Error (NRMSE) [6], and normalized Gini [25], within the target domain. The normalized Gini is computed as the ratio of the Gini coefficient of LTV prediction to the Gini coefficient of LTV labels. This metric evaluates the model's capacity to distinguish high-consumption consumers from the entire consumer base. Lower quantitative values for NMAE and NRMSE, along with higher quantitative values for normalized Gini, signify a stronger predictive performance of the LTV prediction model.

*5.1.3 Baselines.* We compare our model with the following three categories of baselines.

(1) Time Series Forecasting (TSF) Methods:

- ARIMA [4] is one of the most popular linear models for time series forecasting. It has nice statistical properties and great flexibility.
- LSTM [12] is a deep learning model that belongs to a recurrent neural network, which is widely applied in many scenarios for its capable of capturing complex non-linear patterns in time series data.
- Informer [31] is a variant of the conventional Transformer architecture. It mitigates quadratic time complexity and memory usage challenges by introducing probSparse self-attention and distillation operations.
- TCN [2] is a type of convolutional neural network (CNN) designed for sequence modeling. It employs convolutional layers with dilated convolution to capture temporal patterns at different scales.

(2) LTV Prediction Methods:

- Two stage Xgboost [6] involves the classification of customer types followed by the prediction of monetary revenue targeting valuable customers. XGBoost is employed to model both of these essential tasks.
- TSUR [26] proposes a temporal-structural user representation model for LTV prediction, enhancing both temporal and structural encoding. Additionally, it introduces a novel cluster-alignment regularization method to harmonize these two representation types.

(3) CDR Methods:

- DASL [17] investigates domain-specific knowledge extracted from historical sequences and employs a dual learning mechanism to facilitate knowledge transfer between diverse domains.
- PTUPCDR [32] utilizes personalized bridge functions, informed by users' characteristic embeddings, to enable the personalized transfer of knowledge learned from sequences in the source domain.

*5.1.4 Experiment Settings.* We implement our framework and the baselines using PyTorch. For each task and method, we use Adam [15] as the optimizer and set the initial learning rate as 0.001. Regarding model parameters, we have chosen an embedding size of 128 for the hidden layer and 80 for the alignment space, while the number of experts $k$ is set to 4. We set $\{\lambda_1 = 10, \lambda_2 = 20, \lambda_3 = 2, \lambda_4 = 1\}$ as the initial loss factor, and they will drop to 0.01 times after 20 epochs to avoid interfering with the prediction training process. The batch size for all methods is empirically set to 256. We tune all model parameters by fitting them to the validation set and apply an early-stop strategy with a patience of 7 epochs.

Following [20], we evaluate the performance of CDLtvS in cross-domain LTV prediction by randomly excluding historical consumption data in the target domain for a subset of customers, resulting in a group denoted as $C_{na}$ with limited historical information, during the training, validation, and testing phases. The remaining group, endowed with ample data, is represented as $C_a$. Specifically, in our experiments, we vary the proportions of $C_{na}$, represented by $P$, at levels of 0%, 20%, 50%, and 80% of the total customer base. The greater the value of $P$, the higher the proportion of sparse data, which implies more challenging tasks. For the experiments in section 5.3, we assess the model's performance on $C_{na}$ with sparse historical data and $C_a$ with ample historical information.

## 5.2 Performance Comparison

This section presents the results of all the baselines and our model on four LTV prediction tasks under different values of $P$ in Table 1, the best and the second best results are marked in bold and underline. From the experimental results, we have several findings:

- Our results demonstrate superior performance compared to baseline methods. Our method decreases the normalized mean average error by 2.3%, 11.5%, 16.3%, and 14.4%, the normalized rooted mean square error by 6.5%, 12.6%, 10.9%, and 13.2%, while maintaining a high GINI value not less than 0.9442 under various data sparsity levels (0% to 80%).
- Time series forecasting (TSF) methods only use the historical consumption series, and its performance is not satisfactory, especially in tasks with a large $P$ value. Compared with TSF methods, LTV methods make use of customer portrait features, thus achieving better results in sparse data scenes ($P$=80%). Therefore, customer portrait feature data can supplement certain information to enhance the LTV prediction when the historical consumption data of customers is sparse.
- Cross-Domain Recommendation (CDR) methods use the auxiliary data by combining the data from different domains into a single domain and achieve the best performance on the baseline in the scenario of sparse customers. We adapt DASL and

**Table 1: Performance comparison of four LTV prediction tasks. Best and the second best results are marked in bold and underline.**

| | P | 0% | | | 20% | | | 50% | | | 80% | | |
|---|---|---|---|---|---|---|---|---|---|---|---|---|---|
| | Metric | NMAE | NRMSE | GINI | NMAE | NRMSE | GINI | NMAE | NRMSE | GINI | NMAE | NRMSE | GINI |
| TSF | ARIMA | 0.3044 | 2.1839 | 0.9680 | 0.4518 | 2.9219 | 0.6356 | 0.6491 | 3.5338 | 0.3529 | 0.8642 | 4.2030 | 0.1427 |
| | LSTM | 0.3375 | 2.5407 | 0.9768 | 0.4528 | 2.8585 | 0.8255 | 0.6621 | 3.7185 | 0.6343 | 0.9829 | 4.6217 | 0.1439 |
| | Informer | 0.3461 | 2.6967 | 0.9766 | 0.4090 | 2.5038 | 0.8411 | 0.6234 | 3.3494 | 0.5203 | 0.8524 | 4.3021 | 0.2955 |
| | TCN | 0.2604 | 1.4437 | 0.9781 | 0.3913 | 2.0741 | 0.8310 | 0.6215 | 3.2897 | 0.5177 | 0.8262 | 4.0043 | 0.3014 |
| LTV | Xgboost | 0.2754 | 1.4069 | 0.9751 | 0.4783 | 2.0518 | 0.9173 | 0.6257 | 2.5386 | 0.8933 | 0.7079 | 2.8017 | 0.8760 |
| | TSUR | 0.3068 | 1.5763 | 0.9721 | 0.4254 | 2.1288 | 0.8615 | 0.6207 | 3.2052 | 0.7432 | 0.7948 | 3.8653 | 0.6240 |
| CDR | DASL | 0.2639 | 1.4749 | **0.9783** | 0.3287 | 1.6103 | **0.9646** | 0.4038 | 1.8039 | 0.9424 | 0.4898 | 2.0181 | 0.9273 |
| | PTUPCDR | 0.2594 | 1.4707 | 0.9782 | 0.3897 | 2.0781 | 0.8350 | 0.5973 | 3.2035 | 0.5529 | 0.4143 | 1.8081 | 0.8828 |
| Ours | LtvS | 0.2547 | 1.4011 | 0.9775 | 0.3779 | 1.9766 | 0.8681 | 0.5987 | 3.1966 | 0.7081 | 0.7813 | 3.8034 | 0.5527 |
| | CDLtvS | **0.2534** | **1.3492** | 0.9762 | **0.2908** | **1.4061** | 0.9621 | **0.3378** | **1.6063** | **0.9535** | **0.3544** | **1.5692** | **0.9442** |

PTUPRCDR, two typical algorithms in CDR tasks, to LTV tasks, using the same encoding methods of sequence features and portrait features as our model, and their cross-domain learning framework. Compared with DASL, which can distinguish sparse data samples from non-sparse samples by dual learning, the performance of PTUPCDR is seriously degraded in tasks P equals 20% and 50%. This shows that the knowledge transferred from the target domain can not be well utilized, in the case of confusing sparse data and non-sparse data.

- Among all single-domain methods, our method LtvS achieves the best performance in most tasks. Among all the cross-domain methods, our cross-domain method CDLtvS achieves the best performance in most metrics. This demonstrates that our model is effective in LTV prediction in a variety of scenarios with different proportions of sparse data.

## 5.3 Sparse and Non-sparse Data Experiments

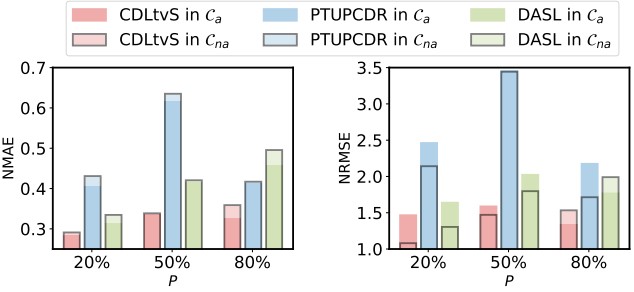

**Figure 4: Comparison of NMAE and NRMSE in sparse and non-sparse data experiments for CDLtvS, PTUPCDR, and DASL at varied sparse data proportions $P$.**

In this section, we distinguish and analyze the performance of cross-domain methods on sparse data sets $C_{na}$ and non-sparse data sets $C_a$ in different tasks. From the results shown in Figure 4, we have the following observations:

- Our CDLtvS model consistently outperforms baseline models on both sparse and non-sparse datasets, providing strong evidence of its broad applicability.
- Regardless of the proportion $P$ of sparse data, all models exhibit superior NMAE performance when dealing with non-sparse data $C_a$ compared to sparse data $C_{na}$. This observation underscores the beneficial impact of ample target domain information on LTV prediction.
- However, in terms of NRMSE performance, the sparse dataset $C_{na}$ outperforms $C_a$ in certain tasks (i.e., $P = 20\%$), indicating potential underfitting of $C_{na}$ in these instances. As the value of $P$ increases, and the proportion of $C_{na}$ within the dataset grows, the model tends to prioritize $C_{na}$ over $C_a$. This results in a relative increase in NRMSE of $C_{na}$, as observed in the results for CDLtvS and DASL when $P = 80\%$.

## 5.4 Ablation Study

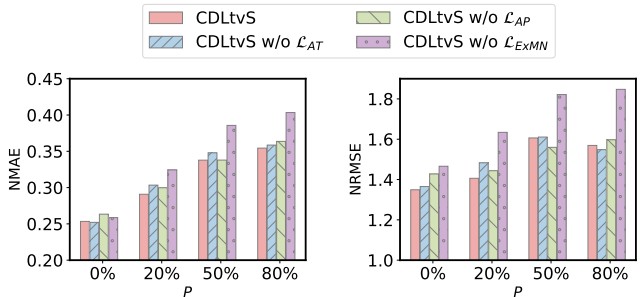

**Figure 5: Comparison of NMAE and NRMSE in ablation studies at varied sparse data proportions $P$.**

In this section, we further compare CDLtvS with several ablation variants to demonstrate the effectiveness and advancement of different sub-modules. Three variants of our approach are compared, including (i) CDLtvS w/o $\mathcal{L}_{AP}$: No align loss $\mathcal{L}_{AP}$ for paired instance, defined in equation 15. (ii) CDLtvS w/o $\mathcal{L}_{AT}$: No align

loss $\mathcal{L}_{AT}$ used to distinguish between sparse and non-sparse data, defined in equation 16. (iii) CDLtvS w/o $\mathcal{L}_{ExMN}$: This variant pre-trains the source and target representation modules without loss $\mathcal{L}_{ExMN}$ defined in equation 6.

In Figure 5, we can see that the loss $\mathcal{L}_{ExMN}$ contributes the most to the final performance since it addresses the long-tail distribution problem in LTV prediction. While loss $\mathcal{L}_{ExMN}$ plays a pivotal role in the first feature representation stage of both the source and target domains, its influence persists in the second stage of cross-domain LTV prediction due to its realization of different feature space for different LTV levels. Simultaneously, the role of loss $\mathcal{L}_{AT}$ is prominent in tasks $P > 0\%$, aligning with our intended design of the model constraint to differentiate between sparse and non-sparse data modeling. And loss $\mathcal{L}_{AP}$ calculated on non-sparse data can be better learned with a large proportion of non-sparse data, thus bringing more improvements to the model in tasks with small $P$.

## 5.5 Fine-tune Performance

This section examines the robustness of our model, analyzes the influence of expert numbers on model performance, and discusses the model evolution in two stages of training.

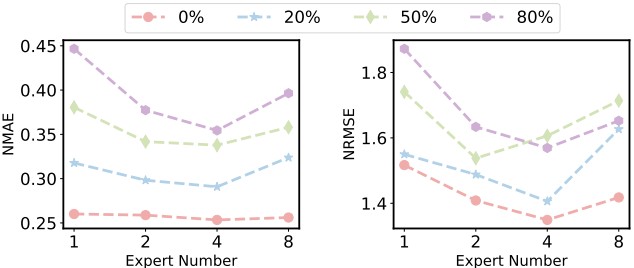

**Figure 6: Comparison of NMAE and NRMSE in experts number experiments. Each line represents a different value of sparse data proportions $P$.**

*5.5.1 The number of experts $k$.* Our model adopts an expert mask network (ExMN) to address the challenge brought by long-tail distribution. The number of experts $k$ will affect the model performance. We vary $k$ in the set $\{1, 2, 4, 8\}$. It can be observed from Figure 6 that $k = 4$ achieves the best performance for our model.

While utilizing numerous experts to customize feature space for customers across different value levels can help alleviate the impact of high losses from high-value customers on low-value ones, the relationship between the number of experts and model effectiveness is not always positively correlated. We identify two key factors. First, when the number of experts becomes excessive, it results in reduced sample assignments to each expert, leading to data insufficiency. Second, our approach involves ranking customer value within a batch, evenly segmenting customer value levels to assign level labels. Increasing the number of experts within a fixed batch size can lead to a decrease in the accuracy of customer LTV level labels obtained through this method.

*5.5.2 Two stage discussion.* We analyze the predicted LTV against the actual LTV of customers in the target domain during Stages 1 and 2, an experimental configuration featuring four experts ($k = 4$)

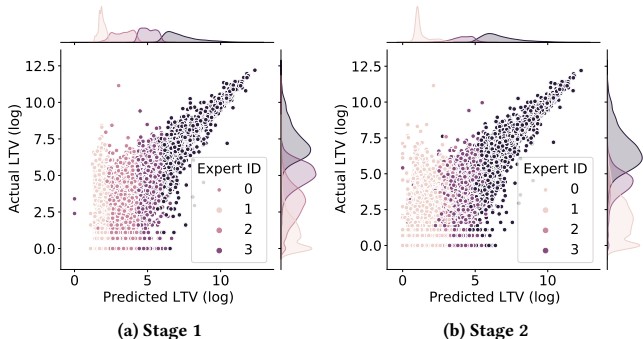

(a) Stage 1                    (b) Stage 2

**Figure 7: Predicted LTV against actual LTV in Stages 1 and 2. We employed distinct colors to differentiate samples assigned to various experts. Within each subgraph, the top and right density plots illustrate the similarity in the distribution of predictions and actual LTV within a single expert, displayed on a logarithmic scale. The central plot demonstrates the effective distinction made by experts among customers in different value levels.**

and a sparse data proportion of $P = 0\%$. As introduced in section 4.2.3, the loss function $L_{ExMN}$ only constrains model optimization in Stage 1, strictly limiting experts to learn customers of varying value levels, while it doesn't apply in Stage 2. Figure 7 illustrates the change in LTV distribution of different experts from Stage 2 to Stage 1, while the experts' ability to differentiate between customers of different value levels remains. In Stage 2, Expert 0 and Expert 1 from Stage 1 merge into a single expert, indicating that samples assigned to Experts 0 and 1 in Stage 1 exhibit little difference in the original data. The Stage 2 model adaptively combines them to assign samples to experts in a more logical manner. In summary, the two-stage training mechanism offers an effective learning path for the ExMN.

## 6 CONCLUSION

In conclusion, to the best of our knowledge, we are the first to introduce a cross-domain method into customer lifetime value prediction in the supply chain platform. We propose the framework CDLtvS to leverage abundant source domain information to overcome data sparsity issues in customer Lifetime Value prediction. This framework incorporates a well-designed expert mask network module and a cross-domain knowledge alignment module to tackle several challenges: 1) modeling the long-tail LTV distribution, 2) mitigating cross-domain model learning bias stemming from long-tail data, and 3) facilitating effective knowledge transfer for both sparse and non-sparse data. Our extensive experiments on a substantial real-world supply chain platform dataset validate the efficiency of our approach and offer valuable insights into model interpretability.

In future research, we look forward to exploring the integration of customer relationship data and enhancing model interpretability. It is also important to consider scalability, ethical considerations, and extending the CDLtvS to broader domains for further advancements.

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
