# OpenReview forum: "A Cross Domain Method for Customer Lifetime Value Prediction in Supply Chain Platform"
_ACM.org/TheWebConf/2024/Conference — TheWebConf24 Oral_

### Official Review · Reviewer_rk1G · 2023-11-19

**Novelty:** 5
**Technical Quality:** 6

**Review:**

This research paper explores the critical role of Customer Lifetime Value (LTV) in customer relationship management within web-based business and digital economics. It addresses the challenge of accurately predicting LTV, especially in supply chain platforms, which significantly impact resource allocation and customer targeting. The work delves into the complexities of LTV prediction, highlighting issues like data sparsity and long-tail distributions that hinder traditional models. The proposed solution, CDLtvS, introduces a novel method that leverages cross-domain information between e-commerce and logistics service platforms within the supply chain scenario. This approach aims to refine LTV predictions by overcoming challenges related to biased knowledge transfer, long-tail distribution modeling, and effective utilization of cross-domain data.

Pros: Novel Approach: CDLtvS introduces a new method specifically tailored for cross-domain LTV prediction within supply chain platforms, addressing a gap in existing research around reliance on ample historical customer data (which is not available in early stages of customer’s lifecycle). Through extensive experiments on real-world data, CDLtvS showcases significant performance gains compared to baseline methods, demonstrating its effectiveness in mitigating data sparsity issues. Also, the ablation tests are helpful in understanding the impact of specific parts of the system and support the design choices made.

Cons: The approach's complexity might limit its immediate applicability or ease of implementation in different scenarios or industries beyond the supply chain domain. Also, the authors have not given any indication for releasing the dataset collected or implementation for the framework - which might make reproducibility/reusability difficult. Furthermore, the model's reliance on multiple loss functions and sub-modules might make the model more sensitive to changes or perturbations, potentially affecting its robustness when deployed in real-world scenarios or when dealing with different datasets.

--- I have read authors response and satisfied with it ----

**Questions:**

Would the dataset and framework implementation be released publicly?

**Reviewer Confidence:**

3: The reviewer is confident but not certain that the evaluation is correct

**Scope:**

3: The work is somewhat relevant to the Web and to the track, and is of narrow interest to a sub-community

---

### Official Review · Reviewer_HxRu · 2023-11-21

**Novelty:** 5
**Technical Quality:** 5

**Review:**

The paper proposes CDLtvS, a novel Cross-Domain method for customer Lifetime Value (LTV) prediction in Supply Chain Platforms (SCP). The paper addresses the challenge of data sparsity in LTV prediction by leveraging cross-domain information from upstream platforms to enhance predictions in downstream platforms. The proposed method incorporates an Expert Mask Network (ExMN) to model the long-tail distribution of LTV and resolve cross-domain learning model bias. The paper presents comprehensive experiments on real-world data from JD.com, demonstrating the effectiveness of CDLtvS in LTV prediction.

Pros:

1. The paper proposes a novel method that leverages cross-domain information, that can effectively addresses the challenge of data sparsity in LTV prediction. Specifically, the paper proposes a novel method named CDLtvS to effectively model the long-tail distribution of LTV and mitigate the model bias problem in cross-domain learning. It further enhances cross-domain knowledge transfer for both sparse and non-sparse data through a various-level alignment mechanism.

2. Extensive experiments demonstrates promising results in real-world datasets.

Cons:

1. The writing and arrangement of the paper can be refined to provide a clearer introduction of the involved problems and definitions, considering that the paper focus on a very specific domain with a lot of proper nouns.

2. The methods for comparison in the experiments are majorly traditional machine learning algorithms. More state-of-the-art methods in cross-domain recommendation or LTV prediction should be included.

3. Some details require further refinement. Specifically, the symbol $\ddot{e}^s$ in equation 3 does not appear elsewhere in the paper and isn't explained. Perhaps it should be $\dot{e}^s$. According to Table 1, the smallest NRMSE among baseline methods is 1.4069 rather than 1.4437 when P=0%, so the proposed method decreases the NRMSE by 4.1% rather than 6.5% at 681 line.

4. According to Table 1, the proposed method LtvS does not achieve the best performance in most tasks among all single-domain methods.

**Questions:**

1. How does CDLtvS compare to state-of-the-art methods in cross-domain recommendation or LTV prediction? Are there any specific advantages or limitations compared to existing approaches?

2. Are there any limitations or potential drawbacks of CDLtvS that should be considered? Are there any specific scenarios or datasets where CDLtvS may not perform well?

3. In section 5.4, the NRMSE of CDLtvS w/o LAP and CDLtvS w/o LAT is smallest when P = 50% and 80% respectively, rather than CDLtvS. Can you explain the reason?

**Ethics Review Description:**

-

**Reviewer Confidence:**

2: The reviewer is willing to defend the evaluation, but it is likely that the reviewer did not understand parts of the paper

**Scope:**

3: The work is somewhat relevant to the Web and to the track, and is of narrow interest to a sub-community

---

### Official Review · Reviewer_AvgM · 2023-11-22

**Novelty:** 5
**Technical Quality:** 5

**Review:**

This paper proposes a cross-domain approach, named CDLtvS, for predicting customer lifetime value in supply chain platforms. This method utilizes rich source domain information to overcome the data sparse problem in customer lifetime value prediction, and solves multiple challenges with an expert mask network module and a cross-domain knowledge alignment module. These challenges include: 1) modeling long-tail LTV distributions, 2) mitigating cross-domain model learning bias caused by long-tail data, and 3) promoting effective knowledge transfer for sparse and non-sparse data.
2.Strengths and Weaknesses
Strength:
1)	This paper has a clear overall structure, introducing the three modules of the CDLtvS framework and clearly discussing the structure of the framework.
2)	This paper conducts a series of experiments to verify the effectiveness and performance of the CDLtvS framework. The experiment uses a real supply chain platform data set and divides it into two groups: a sparse data group and a rich data group. Experiments were conducted on these two groups separately to evaluate the performance of the model.

Weakness:
1)	The major concern is that the technical contribution is somehow incremental. Using contrastive learning to align representation from two domains is a common practice.
2)	It would be better to add more comparison of the proposed model and other LTV methods, to highlight the technical contribution of this paper.

**Questions:**

Please see the weakness above.

**Reviewer Confidence:**

3: The reviewer is confident but not certain that the evaluation is correct

**Scope:**

4: The work is relevant to the Web and to the track, and is of broad interest to the community

---

### Official Review · Reviewer_YiDF · 2023-11-22

**Novelty:** 5
**Technical Quality:** 6

**Review:**

This work develops a predictive model to estimate Customer Lifetime Value (LTV). IMO, this is a high-quality paper with clear explanations, visualization, and analysis. I have a few comments that are primarily concerned with improving the quality of already high-quality work.

-- For me, it was hard to follow the paper in the first read because of the jargon. The authors assume the reader is familiar with the Supply Chain Platforms and how they are operated and modeled. If the target audience is folks in e-commerce, this can be a good paper, but it can be hard to follow for a general audience. I also acknowledge that it is hard to keep the balance given the limited space provided. However, the audience for this work is limited, which has nothing to do with the quality of the work; this is just a feature of this work. I could not follow all the notations and check all the math because it took me a very long time to make myself comfortable with the notations, concepts, and definitions in this space that I was unfamiliar with!

Maybe the authors can provide a table with definitions of all the jargon and abbreviations in the appendices to make it more accessible and provide some low-level examples to help the reader visualize what is happening. Figure 1 is helpful but not enough. In Figure 1, I cannot see the independent/dependent variables and how the element in this figure matches up with the input/outputs in Figure 3. Imagine a biologist is reading this paper!

-- Visualizations are very thoughtful, informative, and easy to understand. It is refreshing to see high-quality work. Well done for Figures 4 and 5!

-- One element that is missing is the confidence intervals. Are the claims statistically significant? I can see the point estimations show an improvement compared to the baseline and other models, but are they statistically significant? Estimating the confidence intervals can be an interesting research question. Confidence intervals are not just useful for model comparison. They also provide a measure to estimate risks.

**Questions:**

--  Are the claims statistically significant?

-- What can a general reader who is not interested in e-commerce learn from the proposed method? Does it have potential applications besides what is presented in the paper? A discussion on these can be constructive.

-- Do the authors plan to provide data and code for the community so they can reproduce the results and build on this research?

-- A discussion on who will be harmed from this research, bias (implicit and explicit forms of bias) propagation, and fairness is missing in this paper. By bias, I do not mean the cross-domain model learning bias that is discussed in the paper. What if the populations change over time (covariate shift)? How is the data created? Are there temporal, spatial, and population-level biases inherited in the data?

**Ethics Review Description:**

A discussion on who will harm from this research, bias (implicit and explicit forms of bias) propagation, and fairness is missing in this paper.

**Ethics Review Flag:**

Yes

**Reviewer Confidence:**

2: The reviewer is willing to defend the evaluation, but it is likely that the reviewer did not understand parts of the paper

**Scope:**

3: The work is somewhat relevant to the Web and to the track, and is of narrow interest to a sub-community

---

### Decision · Program_Chairs · 2024-01-22

**Decision:**

Accept (Oral)

**Comment:**

The authors propose a method of using cross-domain knowledge alignment mechanism to estimate customer lifetime value (LTV) for supply chain platforms. The presentation of the paper can be simplified for more general audiences, a sentiment two other reviewers shared. The results are generally strong across the board, though some statistical significance should be tested. I'm not convinced the authors' response is valid without simply doing a statistical test. Paper has some novelty. Author rebuttals answered most concerns from the reviewers.